# Ultrahigh-Throughput Screening of High-β-Xylosidase-Producing *Penicillium piceum* and Investigation of the Novel β-Xylosidase Characteristics

**DOI:** 10.3390/jof8040325

**Published:** 2022-03-22

**Authors:** Zhaokun Zhang, Mingyue Ge, Qi Guo, Yi Jiang, Wendi Jia, Le Gao, Jianhua Hu

**Affiliations:** 1School of Chemical Engineering, Inner Mongolia University of Technology, Hohhot 010051, China; zhangzhk@tib.cas.cn; 2Tianjin Key Laboratory for Industrial BioSystems and Bioprocessing Engineering, Tianjin Institute of Industrial Biotechnology, Chinese Academy of Sciences, No. 32, Xiqi Road, Tianjin Airport Economic Park, Tianjin 300308, China; m18722542836@163.com (M.G.); guoq@tib.cas.cn (Q.G.); jiangyi@tib.cas.cn (Y.J.); jia_wd@tib.cas.cn (W.J.)

**Keywords:** ultrahigh-throughput screening, xylanolytic enzymes, β-D-xylosidase, thermostability, xylo-oligomers, biomass

## Abstract

A droplet-based microfluidic ultrahigh-throughput screening technology has been developed for the selection of high-β-xylosidase-producing *Penicillium piceum* W6 from the atmospheric and room-temperature plasma-mutated library of *P. piceum*. β-xylosidase hyperproducers filamentous fungi, *P. piceum* W6, exhibited an increase in β-xylosidase activity by 7.1-fold. A novel β-D-xylosidase was purified from the extracellular proteins of *P. piceum* W6 and designated as PpBXL. The optimal pH and temperature of PpBXL were 4.0 and 70 °C, respectively. PpBXL had high stability an acidic pH range of 3.0–5.0 and exhibited good thermostability with a thermal denaturation half-life of 10 days at 70 °C. Moreover, PpBXL showed the bifunctional activities of α-L-arabinofuranosidase and β-xylosidase. Supplementation with low-dose PpBXL (100 μg/g substrate) improved the yields of glucose and xylose generated from delignified biomass by 36–45%. The synergism between PpBXL and lignocellulolytic enzymes enhanced delignified biomass saccharification, increased the Xyl/Ara ratio, and decreased the strength of hydrogen bonds.

## 1. Introduction

Lignocellulosic biomass is the most abundant natural polysaccharide available on Earth, which can be converted into biofuels to decrease greenhouse gas emission and strengthen energy security [1,2]. Lignocellulosic biomass is consisted of lignin (10–25%), hemicellulose (20–40%) and cellulose (40–60%) [3]. Xylan, the second most abundant renewable resource on Earth, is the main component of hemicellulose [4]. The complete degradation of xylan requires various xylanolytic enzymes, including acetylxylan esterase (EC 3.1.1.72), α-glucuronidase (EC 3.2.1.139), α-L-arabinofuranosidase (EC 3.2.1.55), β-xylosidase (EC 3.2.1.37) and endoxylanase (EC 3.2.1.8). The β-1,4-glycoside bonds in xylan can be hydrolyzed by xylanase into xylobiose and xylo-oligosaccharides [5]. However, xylo-oligomers greatly suppress hemicellulase activity with increasing concentrations. β-xylosidase is the main enzyme that catalyzes the hydrolysis of short chain xylooligosaccharide from the non-reducing ends of β-1,4-linked D-xylose residues to secrete xylose as the final product. Therefore, β-xylosidases play a critical role because they have the ability to cleave glycosidic bonds, and ameliorate product inhibition of xylanases for hemicellulase activity [6,7].

β-xylosidases are categorized into different glycoside hydrolase families (GH1, GH3, GH30, GH39, GH43, GH51, GH52, GH54, GH116, and GH120, according to the GH classification system CAZy (Carbohydrate active enzymes database—http://www.cazy.org, accessed on March 2022) [8]. β-xylosidases contain a complex group of enzymes for improving the nutritional values of grain silage and agricultural feed, as well as biomass saccharification [9,10,11]. A variety of underutilized agricultural residues and lignocellulosic waste can be used as low-cost feedstocks for biofuel production. *A. fumigatus* XC6 β-xylosidase can promote ethanol production to 80% of the theoretical value [9]. There is an example of β-xylosidase application in the strawberry ripening and pectin solubility [12]. These varied applications require thermostable β-xylosidases because of their stability at high temperatures over a long period of time [13]. Some β-xylosidases from *Aspergillus*, including *A. fumigatus*, *A. brasiliensis* and *A. niger*, exhibit optimal activities at high temperatures (55–75 °C) [14]. However, the application of β-xylosidases is limited by thermal stability. Therefore, novel β-xylosidases with unique characteristics should be explored for industrial application.

Microbial β-xylosidases occur in bacteria and fungi, with those of the latter group being the most studied [8]. At present, β-xylosidases are primarily produced by ascomycete fungi in industry [15]. As β-xylosidases could secreted to fermentation broths at high levels, which are suitable for subsequent collection process. Although many filamentous fungi can produce β-xylosidases, an approach in high-throughput screening for high-β-xylosidase-producing filamentous fungi remains unavailable. The isolation of high-β-xylosidase-producing mutants from a mutant library requires a long screening cycle, including clonies separation in wells of microtiter plates and determination of enzyme activity [16]. However, the construction of an ultrahigh-throughput screening system for *Penicillium piceum* is challenging given the filamentous nature of this fungus and the secretion of β-xylosidase extracellularly. The development of the ultrahigh-throughput screening system for β-xylosidase considerably increased the chance of obtaining desired properties, which would improve strain screening efficiency and shorten strain screening cycle.

In this paper, a new high-throughput screening method for β-xylosidase producing filamentous fungi is proposed. The characteristics and exact role of this novel β-xylosidase in biomass degradation are demonstrated. Subsequent characterization and application reveal the novel β-xylosidase’s unique properties, making it valuable in the potential industrial applications of biomass degradation.

## 2. Materials and Methods

### 2.1. Microorganism and Culture Conditions

*P. piceum* (China General Microbiological Culture Collection Center; Beijing, China; CGMCC 8339) was a β-glucosidase hyperproducing strain obtained by dimethyl sulphate mutagenesis [17]. The medium (g/L) used consisted of 3.3% corncob steep liquor, 2.7% Avicel, 0.5% (NH_4_)_2_SO_4_, 0.6% KH_2_PO_4_, 0.1% MgSO_4_, 0.2% Tween-80 and 0.25% CaCO_3_. After inoculation into 50 mL of the medium in a 300-mL conical flask, the spore suspension was grown at 28 °C for 5 days under rotatory shaking at 180 rpm [18].

*T. reesei* A2H (China General Microbiological Culture Collection Center; Beijing, China; CGMCC; 21470) was a cellulase-hyperproducing strain generated from ARTP mutagenesis. For strain recovery, mycelia agar disks were inoculated on fresh PDA and cultivated at 30 °C for 7 days until formation of conidia. Spore suspension (1 mL) was added into 30 mL preculture (10 g/L corn steep powder, 10 g/L glucose, pH 5.0) in a 250-mL flask and incubated at 28 °C for 24 h at 180 rpm. After transferring 5% precultured inoculum size to 30 mL of production medium in a 250-mL flask at 26 °C for 5 days at 180 rpm. The medium (g/L) consisted of 1.7% corn steep powder, 0.6% KH_2_PO_4_, 0.5%(NH_4_)_2_SO_4_, 0.25% CaCO_3_, 0.1% MgSO_4_∙7H_2_O, 0.2% Tween 80, and 3% inducer and had an initial pH of 5.0.

### 2.2. Atmospheric- and Room-Temperature Plasma (ARTP) Mutation

After culturing for 5 days, the spores were harvested with sterile distilled/deionized water and filtered through 0.22 µm. After diluting the spore suspension to 10^7^ spores/mL, 10 μL of diluted suspension was spread onto sterilized steel dishes, followed by exposure to a helium gas flow for 180 s. The plasma reactor was kept at a distance of 2 mm. Subsequently, the suspension was placed in 1 mL distilled water. After ARTP treatment, the spores were precultured at 30 °C before droplet encapsulation.

### 2.3. Droplet Production and Microfluidic Testing

To produce droplets, a suspension of 10^6^ spores/mL in medium containing 100 µM β-xylosidase fluorogenic substrate 4-methylumbelliferyl-β-xylobioside (4MUX) and a surfactant-containing fluorinated oil were used as the aqueous and oil phases, respectively [19]. The flow rates were 650 and 1000 μL/h for the oil and aqueous phases, respectively. The commercial microfluidic drop-making device was used to generate the uniform droplets. The droplets were harvested using a syringe (2 mL; BD Biosciences, USA) and incubated at 28 °C for specific period. Finally, the droplets were analyzed by the microfluidic sorting device at the optimal flow rates of 1000 and 20 μL/h for spacing oil and droplets (HFE-7500, 3M, Minneapolis, MN, USA), respectively.

The content of β-xylosidase in droplets was assessed by using the fluorogenic substrate 4MUX (Megazyme, Wicklow, Ireland) at excitation and emission wavelengths of 365 and 450 nm, respectively. A focused laser was used to excite the droplets, and the obtained fluorescence intensities were determined using the microfluidic system as described previously [20]. The high β-xylobioside activity, which reflected by strong fluorescence intensities, were subjected to further analysis.

### 2.4. Purification of β-Xylosidase from P. piceum

After ultrafiltration with a *M*r 10,000 cut-off Millipore membrane (Millipore, Hercules, CA, USA), the supernatant of *P. piceum* was loaded onto an anion exchange SP column (GE Healthcare, Uppsala, Sweden). Based on the three peaks, the fractions were collected and subjected to further separation. After loading onto an anion exchange DEAE column (20 and 1.6 cm, GE Healthcare, Uppsala, Sweden), the combined fractions were equilibrated with sodium acetate buffer (20 mM, pH 5.0). Then, elution was performed in the same buffer by a linear NaCl gradient (0–1 M) at 2 mL/min, and 1 mL fraction was collected.

### 2.5. Matrix-Assisted Laser Desorption/Ionization Time-of-Flight Mass Spectrometry

Internal amino acid sequence analysis of proteins was conducted via the in-gel digestion. Mass spectrometry was used to sequence different peptides at the Tianjin Institute of Industrial Biotechnology, CAS. The simulated three-dimensional structure of the β-xylosidase was obtained by SWISS-MODEL (https://swissmodel.expasy.org/, accessed on January 2022).

### 2.6. Determination of Protein Concentration

A trace protein assay kit was utilized to measure protein concentrations. Specifically, the bicinchoninic acid method was employed (Vazyme Biotech Co., Ltd., Nanjing, China) [21].

### 2.7. Determination of Xylose and Arabinose Concentration

Xylose and arabinose levels were detected on an Aminex HPX-87H column (BioRad, Hercules, CA, USA) by high-performance liquid chromatography (HPLC) with a refractive index detector (Shimadzu, Kyoto, Japan). The mobile phase, flow rate and temperature were 5 mM H_2_SO_4_, 0.6 mL/min and 60 °C, respectively [22].

### 2.8. Biochemical Characterization of Purified Protein

#### 2.8.1. Enzyme Assay and Substrate Specificity

The β-xylosidase activity was routinely measured through the amount of *p*-nitrophenol released from *p*-nitrophenyl-β-D-xylopyranoside (*p*NPX). The assay mixture contained 20 µL diluted enzyme and 80 µL *p*NPX in a working volume of 100 µL. The reaction was carried out at 70 °C for 10 min and then stopped by 100 µL sodium carbonate. The *p*-nitrophenol released from *p*NPX was determined by the absorbance at 410 nm. The amount of enzyme that produced 1 µmol of *p*-nitrophenol per minute was denoted as 1 unit of the enzyme activity. The assay was carried out as described previously [23]. The specificity of purified β-xylosidase against different substrates was investigated using 1% *p*-nitrophenyl-α-L-arabinofuranoside (*p*NPAF), *p*-nitrophenyl-β-D-*p*-nitrophenyl-β-D-glucose (*p*NPG), *o*-nitrophenyl-β-D-xylopyranoside (*o*NPX) and birch wood xylan. In certain assays, *p*NPX was substituted with *o*NPX, *p*NPG or *p*NPAF.

#### 2.8.2. Effect of Temperature

The optimal temperature for purified β-xylosidase was determined in phosphate buffer (pH 4.0) and 50 mM citrate at 30–70 °C for 30 min. Thermostability of the enzymes was assessed after incubation at different temperature (30–70 °C) by using the same buffer with no substrate. Sampling was conducted at different time points, and the residual activities were evaluated.

#### 2.8.3. Effect of pH

The optimal pH for purified β-xylosidase was determined in phosphate buffer and 50 mM sodium citrate at 70 °C and pH 2.0–8.0 for 30 min. The pH stability of the enzymes was evaluated by detecting the residual activities at 24-h intervals.

#### 2.8.4. Effect of Metal Ions

The effect of metal ions on enzyme activity was assessed with 1 mM final concentration of the substances. The activity of the enzymes with no metal ions was regarded as 100%.

### 2.9. Synergism Experiments

Synergism assays were conducted in duplicate trials in a 100-mL centrifuge tube. The reaction mixture consisted of 5% (*w*/*v*) delignified biomass (corn stover/corn cob) and 10 FPU cellulase per gram substrate in a working volume of 50 mL. Delignified biomass were acquired after alkali pretreatment. Lignocellulolytic enzymatic preparation from *T. reesei* A2H was a cocktail of cellulase and hemicellulase. Lignocellulolytic enzymes from *Trichoderma reesei* were cultivated with filter paper for enzyme activity of 30 IU/mL and xylanase activity of 400 IU/mL. Then, 100 μg purified protein/g substrate was used as the supplementation dosage. Saccharification was carried out at 50 °C for 3 days. Glucose and xylose in enzymatic hydrolysate were measured according to the method [18].

### 2.10. Hydrogen Bond Intensity (HBI) Analysis

A vacuum dryer (FD-IC-50, Beijing, China) was used to dry the biomass specimens at −20 °C for 24 h. IR spectral data were obtained using an IR spectrophotometer (FTIR 710; Nicolet, Thermo Fisher Scientific, Waltham, MA, USA). The HBI was calculated in accordance with the method [18].

### 2.11. Statistical Analysis

To compare DDGS composition with or without lignocellulolytic enzymes, ANOVA was employed, followed by Tukey’s multiple comparison post-test. Statistical tests were conducted with SigmaPlot V11 [24]. Level of statistical significance was set at *p* < 0.01 [25].

## 3. Results and Discussion

### 3.1. Spore Germination and Droplet Fluorescence Intensity

The enzymes from *A**. niger* with no β-xylosidase activity was detected using this droplet-based microfluidic method as a negative control. There was no fluorescence signal released from β-xylosidase fluorogenic substrate (4MUX) in the *A. niger* droplets (Appendix A). There were evident fluorescence signals released from β-xylosidase fluorogenic substrate in the *P. piceum* droplets (Appendix A). During the droplet cultivation, the hyphal tips of *P. piceum* grew rapidly after spore germination. Simultaneously, the β-xylosidase activity detected increased with hyphal growth. Droplets exhibiting high levels of fluorescence was consistent with high-β-xylobioside activity of *P. piceum.* A positive linear relationship was observed between fluorescence intensity released from 4MUX and β-xylosidase activity (Figure 1A). The fluorescence intensity and hyphal morphology were considered as important parameters for incubation time of droplets. After 14 h cultivation, spores of *P. piceum* were germinated, and the hyphae were extended until reaching the edges of the droplets at 22 h. If the culture time continued to 24 h, long mycelium resulted in punctured droplets (Figure 1B). Observing the 22 h culture indicated the presence of fluorescence intensities and hyphae were not extended beyond the droplets. The incubation time of *P. piceum* droplet was determined as 22 h. The results indicated that β-xylosidase-producing strains could be screened by droplet-based microfluidic approach.

### 3.2. Screening from Library Mutagenesis and Shake-Flask Fermentation Tests

After ARTP mutation, the droplet-wrapped hypha of *P. piceum* was analyzed by fluorescence microscopy followed microfluidic sorting at 10,000 variants per hour. After demulsification, the sorted spores-containing droplets were spread onto PDA for validation based on high fluorescence intensity. The randomly picked 20 colonies were inoculated into the fermentation medium, followed by β-xylosidase activity determination. These colonies had an increase in β-xylosidase activity compared the parent strain *P. piceum* H16 (Figure 2). One W6 mutant showed the highest β-xylosidase activity against *p*NPX of 171.2 IU/mL, which was about 7.11-fold higher than *P. piceum* H16. This result shows that β-xylosidase-producing strains can be screened through the droplet-based microfluidic system.

### 3.3. Purification of β-Xylosidase from P. piceum

The concentrated culture filtrate from *P. piceum* W6 was loaded onto the anion exchange SP column, resulting in three peaks denoted as peak I, II and III (Figure 3A). The most of β-xylosidase activity in *P. piceum* W6 supernatant showed in the peak II (data not shown). Therefore, the second peak was concentrated and loaded onto the DEAE column and eluted with 0–1 M NaCl (Figure 3B). The purified protein was determined as a single band in SDS-PAGE. The molecular weight of purified protein was 65 kDa (Figure 3C).

### 3.4. Identification of β-Xylosidase from P. piceum by Internal Peptide Sequences

The single band from SDS-PAGE was hydrolyzed by trypsin digestion to generate six major peptide fragments. HPLC-ESI-MS/MS results showed that KVSVVVDASSK (peptide 1), KELGSLGKAQTYFR (peptide 2), KSYERWGK (peptide 3), KGVPLDLISFHAK (peptide 4), RVLSTQGIDK (peptide 5) and RVDNKHSNSYAK (peptide 6) (Appendix A) were obtained. The sequences of the purified protein were completely similar with those of β-xylosidase in *P. piceum*-sequenced genome (unpublished data). The purified protein was identified as PpBXL. The protein sequence of PpBXL was compared against with publicly available sequence data through BLAST. The protein belonged to the glycoside hydrolase family 39, having 57.25% similarity with the 1.4-β-xylosidase from *Rhizodiscina lignyota*, 57.40% similarity with β-xylosidase from *Rasamsonia emersonii*, 56.20% similarity with β-xylosidase from *Oidiodendron maius* and 55.16% similarity with β-xylosidase from *Viridothelium virens*. The sequences of PpBXL have been uploaded to the NCBI database (GenBank accession number: OM055664).

### 3.5. The Structure of β-Xylosidase from P. piceum

The three-dimensional structure of PpBXL, belonging to GH39, was predicted using β-xylosidase from *Thermoanaerobacterium saccharolyticum* (PDB ID code: 1UHV) [26] as template. The PpBXL gene was 1722 bp long, encoding a polypeptide of 574 amino acids. The molecular mass of purified PpBXL was 65 kDa, suggesting it had a monomer structure. Many types of multimers, such as homotetramers, homotrimers and homodimers, have been detected in β-xylosidases [27]. A few works reported that quaternary structure was important for enzymatic activity [28]. Our result was different from the previous view, suggesting that quaternary structure was not necessary for enzymatic activity of β-xylosidase. The structure of PpBXL was relatively conserved and mainly consisted of 11 α-helices, 16 β-stands and six 3_10_-helices a’ to f’ (Figure 4). The catalytic domain contained a β-hairpin and a canonical (β/α)_8_ protruding from barrel. The upper side of barrel contained an active-site pocket that can bind to D-xylose.

### 3.6. Substrate Specificity of β-Xylosidase from P. piceum

As shown in Table 1, PpBXL could hydrolyze *p*NPX and *o*NPX but more favorable towards *p*NPX. PpBXL showed specific activities against *p*NPX and *o*NPX, i.e., 320.5 and 79.3 IU/mg, which were higher than that of β-xylosidase from *Caulobacter crescentus* (215 IU/mg) or *Geobacillus sp*. (133 IU/mg) against *p*NPX [8,13]. Interestingly, PpBXL showed a specific activity against *p*NPAF (107.8 IU/mg), suggesting that PpBXL had arabinofuranosidase activity. PpBXL showed the bifunctional activities of α-L-arabinofuranosidase and β-xylosidase. Most of β-xylosidase shows no arabinofuranosidase activity [5]. *WSUCF1* β-xylosidase was only active against *p*NPX [13]. The bifunctional activities of α-L-arabinofuranosidase and β-xylosidase, e.g., β-xylosidase from *Butyrivibrio fibrisolvens*, *Thermoanaerobacter ethanolicus* and *Sporotrichum thermophile* [29,30,31] have been reported. It has been suggested that these β-xylosidases had both xylosidase and arabinosidase activity. However, the bifunctional activities of other β-xylosidases against *p*NPAF and *p*NPX were relatively lower than that of PpBXL in this paper.

### 3.7. Effects of Metal Ions on PpBXL Activity

Different divalent cations have strong inactivation of enzyme activity, as similar to other fungal β-xylosidases. In this study, Cu^2+^, Hg^2+^, Ag^2+^ and Ni^2+^ exerted strong inhibitory effects on the purified enzymes even at 1 mmol/L (Table 2). Ni^2+^ and Hg^2+^ decreased the enzyme activity by 90%. Ca^2+^ and Mg^2+^ activated the activity by 17.4% and 5.2%, respectively. The metal ions are bound to the carbonyl oxygen atoms of the active site, and the calcium ion may have a stabilizing function [32]. Based on the present study, Ca^2+^ played an important role in activity activation and structural stabilization. β-mercaptoethanol inhibited β-xylosidase activity, suggesting the absence of sulfhydryl groups in the catalytic domain.

### 3.8. pH and Thermal Stability of PpBXL Activity

PpBXL had excellent stability over a wide pH range of 3.0–8.0 and showed an optimum pH of 4.0 (Figure 5A), suggesting that PpBXL had remarkable potential for industrial application in acidic pH. At pH 3.0 and 5.0, PpBXL had 90.7% and 93.6% of its maximum activities, respectively, suggesting that this enzyme was an acid-tolerant β-xylosidase (Figure 5A). PpBXL achieved a good pH stability because >90% of its activity was retained after treatment at 3.0–5.0 for 4 days (Figure 5C). The majority of β-xylosidases were activated at neutral pH values [8,13,14]. The optimal pH 4.0 of PpBXL in this work was lower than that of previously reported β-xylosidases. The pH decreased to 4.0 during the simultaneous saccharification and fermentation (SSF) of lignocellulosic biomass to biofuels. Most hemicellulase and cellulase could retain only 60% of their maximum activities at pH 4.0 [33]. PpBXL can be applied in bioethanol production under acidic condition due to its acid-tolerance characteristics.

The optimal temperature of PpBXL was 70 °C. PpBXL maintained 55.2% and 87.6% of its maximum activities at 80 °C and 60 °C, respectively (Figure 5B). Some β-xylosidases from *Aspergillus* (such as *A. niger*, *A. brasiliensis*, and *A. fumigatus*), *Geobacillus*, and *Caulobacter* strains present optimal activity at 55–75 °C [13,14,32]. The β-xylosidase from *A. fumigatus* XC6 was thermally stable within the range of 60–75 °C, and retained maximum activity at 75 °C for 24 h [9]. However, the high-temperature resistance (>7d) and acid-tolerant characteristics were not observed in the pH profile. Additionally, PpBXL had outstanding thermostability with a thermal denaturation half-life of 10 days at 70 °C (Figure 5D), and this value was increased compared with β-xylosidase from the thermophilic *Geobacillus* strain (9 days at 70 °C) [9].

### 3.9. Effects of PpBXL on the Enzymatic Degradation of Delignified Biomass

PpBXL was added to *T. reesei* lignocellulolytic enzymes at a low dose of 100 μg/g substrate in the saccharification to elucidate the role of PpBXL in improving the enzymatic degradation of delignified biomass (delignified corn stover and delignified corn cob). PpBXL had remarkable synergism with lignocellulytic enzymes from *T. reesei* during the delignified biomass saccharification. Although the supplemental dose of PpBXL was low, the glucose and xylose yields released from delignified corn stover by the *T. reesei* lignocellulolytic enzymes increased by 41.4% and 39.3%, respectively. The glucose and xylose yields released from delignified corn cob after PpBXL supplementation increased by 36.3% and 44.6%, respectively.

The delignified biomass could increase the yields of hemicellulose after alkali pretreatment. However, it remains a challenge to achieve the complete enzymatic degradation of xylan to xylose. Xylanases could hydrolyze xylan into xylose and various xylo-oligosaccharides. As these xylo-oligosaccharides were not able to undergo fermentation, they could be detected in the hydrolysates of lignocellulosic materials, thus resulting in lower yields of final product. Xylaolytic enzymes in most commercial enzyme preparations were not capable to hydrolyze xylan completely, leading to high xylo-oligomer contents retaining in the hydrolysis broth. The accumulation of xylo-oligomer led to decreased initial hydrolysis efficiency of cellulase [34]. β-xylosidase could synergy with xylanase for the complete degradation of hemicellulose. The *WSUCF1* β-xylosidase and xylanase were added to increase xylose yields to 90% during birchwoodxylan hydrolysis, which further enhanced the fermentation product yields [13]. Similar observations were observed in this work, the hydrolysis efficiency of biomass was improved via the decreased inhibitory effect of xylo-oligomers after PpBXL supplementation. The Xyl/Ara ratio negatively affected the degree of Ara substitution of xylan and was positively related to biomass digestibility [19]. Delignified corn stover and corn cob after PpBXL supplementation had considerably higher xylose (Xyl)/arabinose (Ara) ratios (9.12 and 13.23, respectively) than those before supplementation (3.01 and 5.12, respectively) (Table 3). This result showed the hydrolysis efficiency of xylanase was remarkably improved (*p* < 0.01) due to PpBXL supplementation. When PpBXL could enhance the hemicellulose degradation, the hydrogen bonds between hemicellulose and cellulose were simultaneously disrupted. It is found that HBI of delignified biomass decreased (*p* < 0.01) after PpBXL supplementation. Hence, PpBXL supplementation evidently improved the saccharification of biomass.

## 4. Conclusions

A droplet-based microfluidic ultrahigh-throughput screening technology has been developed for the screening of high-β-xylosidase-producing strain, which effectively improve the quality of strain breeding. PpBXL can serve as a promising bifunctional enzyme with high β-xylosidase and α-L-arabinofuranosidase activities. Moreover, PpBXL had excellent thermostability with a thermal denaturation half-life of 10 days at 70 °C. At low concentrations, PpBXL remarkably synergized with lignocellulolytic enzymes from *T. reesei*. PpBXL might improve the hydrolysis efficiency of lignocellulose degradation by decreasing the inhibitory effect of xylo-oligomer. The development of enzyme cocktails by using novel enzyme components with unique characteristics might provide a new way for improving lignocellulose hydrolysis.

## Figures and Tables

**Figure 1 jof-08-00325-f001:**
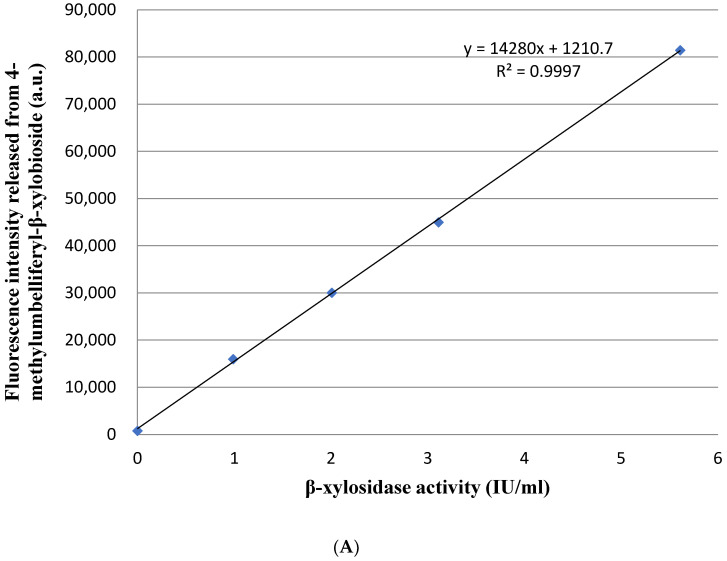
(**A**) Positive linear relationship between fluorescence intensity released from 4-methylumbelliferyl-β-xylobioside and β-xylosidase activity. (**B**) Fungal hyphal growth and fluorescence signal detection in droplets at different cultivation times (scale bar = 50 μm).

**Figure 2 jof-08-00325-f002:**
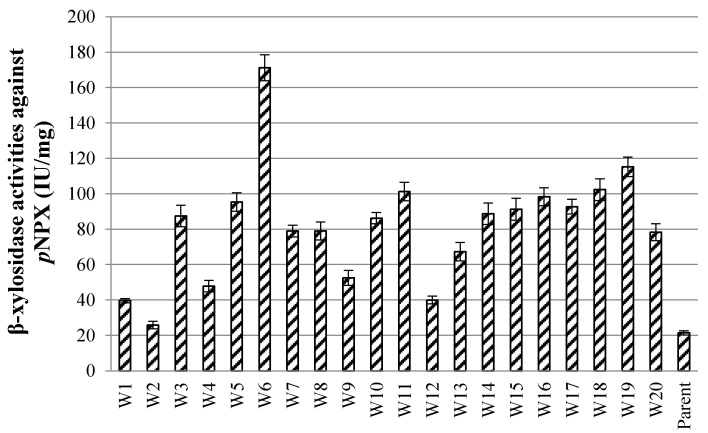
β-xylosidase activity of the screened mutants via the droplet-based microfluidic ultrahigh-throughput screening technology.

**Figure 3 jof-08-00325-f003:**
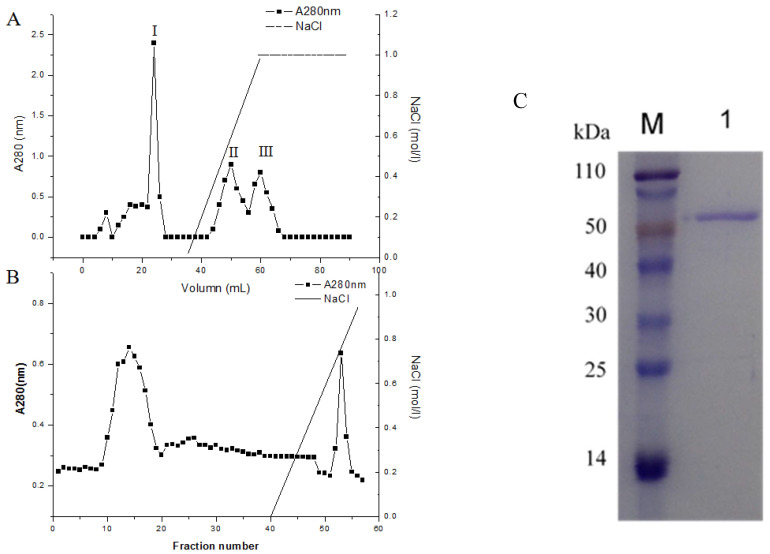
PpBXL purification process: (**A**) SP Fast Flow chromatography, (**B**) DEAE Fast Flow chromatography and (**C**) SDS-PAGE of the purified PpBXL.

**Figure 4 jof-08-00325-f004:**
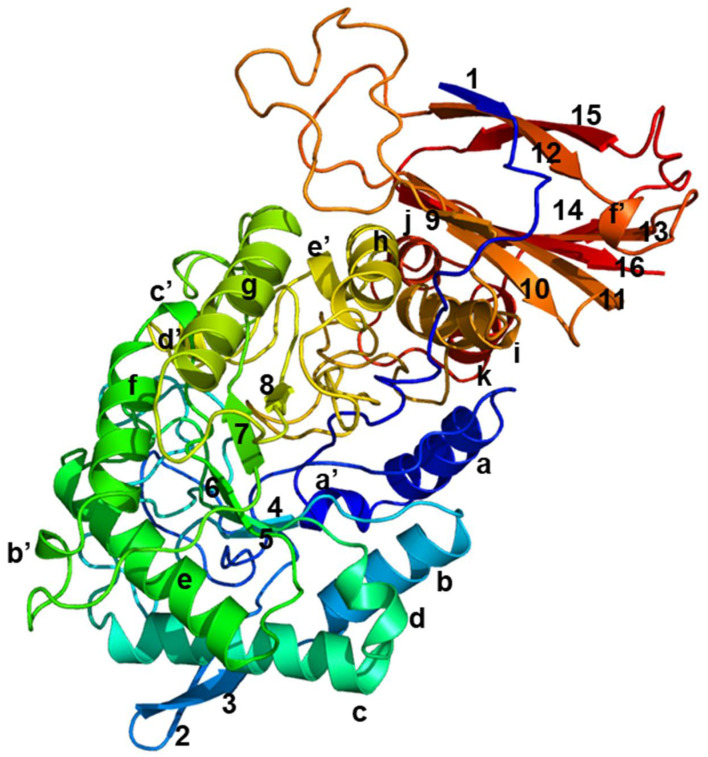
Simulation of three-dimensional structure of β-xylosidase. Eight α-helices are indicated as a to k, sixteen β-strands as 1 to 16, and six 3_1__0_-helices as a’ to f’.

**Figure 5 jof-08-00325-f005:**
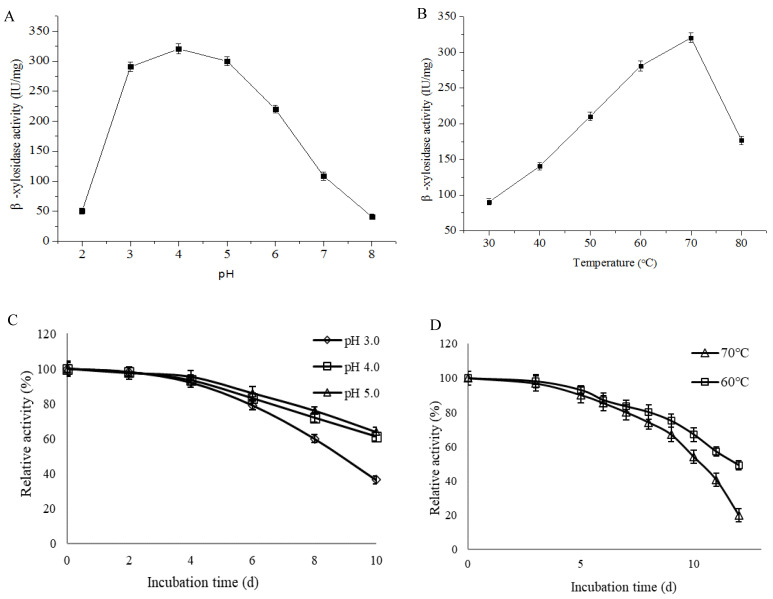
Optimum (**A**) pH and (**B**) temperature of PpBXL. (**C**) pH and (**D**) temperature stabilities of PpBXL.

**Table 1 jof-08-00325-t001:** Substrate specificity of the purified PpBXL.

Substrate	Specific Activity (IU/mg)
*p*NPAF	107.8
*p*NPX	320.5
*o*NPX	79.3
*p*NPG	0
Xylan, birchwood	0

Enzyme activity was measured against different substrates with concentration of 1%.

**Table 2 jof-08-00325-t002:** Effects of some metal ions and reagents on purified PpBXL.

Metal Ions and Reagents	Relative Activity (100%) ^a^
Control	100.0
CaCl_2_	117.4
FeCl_2_	103.2
CuSO_4_	33.4
MgSO_4_	105.2
MnCl_2_	102.2
AgCl_2_	11.4
HgCl_2_	9.8
NiCl_2_	8.2
β-mercaptoethanol	15.3

^a^ Enzyme activity was determined in the presence of 1 mM final concentrations of the substances.

**Table 3 jof-08-00325-t003:** Biomass characteristics and saccharification before and after PpBXL supplementation.

Substrate	Delignified Corn Stover				Delignified Corn Cob			
	Glucose Yield (g/L)	Xylose Yield (g/L)	Xyl/Ara	HBI	Glucose Yield (g/L)	Xylose Yield (g/L)	Xyl/Ara	HBI
After hydrolysis by *T. reesei* cellulase	8.11	6.71	3.01	4.62	11.21	9.48	5.12	2.78
After hydrolysis by *T. reese*i cellulase and PpBXL	11.47	9.35	9.12	3.12	15.28	13.71	13.23	2.01
*p*-value	<0.01	<0.01	<0.01	<0.01	<0.01	<0.01	<0.01	<0.01

## Data Availability

All data generated or analyzed during this study are included in this published article (and its Appendix A).

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
