# Peer review of "Ultrahigh-Throughput Screening of High-β-Xylosidase-Producing Penicillium piceum and Investigation of the Novel β-Xylosidase Characteristics"

_jof, 2022, doi:10.3390/jof8040325_

Round 1

Reviewer 1 Report

Dear Authors,

The comments about your submitted work are pointed below:

The submitted paper “Ultrahigh-throughput screening of high-β-xylosidase producing Penicillium piceum and investigation of the novel β-xylosidase characteristics” by Zhang et al., reports on a droplet-based microfluidic ultrahigh-throughput technique, used for the selection of a P. piceum mutant β-xylosidase hyper producer.

In my opinion, the manuscript interestingly introduce the droplet-based microfluidic ultrahigh-throughput technique to select a mutant that hyper produce a β-xylosidase activity with interesting characteristics. Moreover the enzyme showed a good applicability in biomasses degradation.

The manuscript could be implemented in the methods section describing more clearly the experimental procedure.

In the Introduction, I suggest to consider adding some other examples of characterized beta xylosidases and the potential biotechnological applications of the enzyme.

In my opinion, the Results and Discussion could be improved by implementing the section with comparisons of previous literature, which of course deals with the same subject.

Tables and Figures need to be improved.

Some points need to be clarified and justified are listed below:

Line 44-46 - Please may you explain better this concept?

As is well known, β-xylosidase is the major hydrolytic enzyme that degrades the non-reducing ends of β-1,4-linked D-xylose residues to release xylose as the final product from short xylooligosaccharide.

Line 67 – Please, may you point to some of the potential applications of the enzyme?

Line 85 -In the materials and methods section, there are few descriptions of the experiments and many references to previous works. This is certainly legitimate, but in my opinion, it makes the reading and immediate understanding of the execution of the experiments not very fluid. I suggest, for the most important methodologies, for example for the method of detecting the droplet in the microfluidic system, to report the experimental procedure.

Line 87 – Please correct verification.

Line 99 -105: The definition of β-xylosidase units is the same as reported in Iembo et al. Please, for a better understanding of the text, report the definition.

Line 103- Please specify the name of the pNPG substrate.

Line 111 - Please, may you specify in the Materials section at what interval of time the samples were withdraw?

Line 129-130 – The tube used for the experiment was 10 mL centrifuge tube, but the working (please correct the misprint into the text) was 50 mL. Please verify the volumes.

Line 130 – How were the biomasses delignified?

Line 184 -190 -192 –Please, pay attention to P piceum, it must be written in italics.

Line 193 - Please, correct screened.

Line 234 – Have you considered to determine the molecular weight of the native protein? This can confirm the tetrameric structure of the enzyme.

Table 1 – 2- The character of the table are very big. If it is not a specific editorial request please reduce the characters. Moreover, specify the final concentrations of both substrates and metals tested, in order to make the text easier to understand.

Line 265 – The Nickel is missing in the table 2.

Line 271 - Please explain the concept better. Is it therefore understood that disulfide bonds may be present in the catalytic site of the enzyme?

Line 281 – Please, could you give some examples of industrial application of β-xylosidases at acidic pH.

Line 304 - The β xylosidase enzyme works in combination with xylanases for the complete degradation of the hemicellulosic matrices. When it is pointed out that β-xylosidase works in combination with cellulases from T. reesei, it is perhaps meant that the enzymatic cocktail, consisting of numerous enzymes (cellulase, β-glucosidase, xylanase, arabinofuranosidase, etc.) supplemented with the enzyme studied improves the final monosaccharides yield? Please make the concept clearer and more explicit.

Line 308- Please specify at least the first time what is meant by Xyl, maybe xylose I suppose.

Figure 5: Please, be careful to uniform the characters used in the figures. Figure 5 shows the ordinate bar also for the graph concerning the temperature. Furthermore, the two graphics are staggered and with a bad resolution. Also be careful to report the ordinate bar for graph 5 d. Furthermore, in the legend figure 5 consists of 4 graphs named a b c and d, while the wording is highlighted only for graphs a and c.

Table 3- As for tabs 1 and 2 the graphic characters are very large, pay attention to the size of the boxes and make them uniform. Some numbers and letters are not in the correct position.

Reviewer 2 Report

The subject of the study is of great interest  for choosing xylanase which a potent industrial applications and the effort being performed in the study is clear starting from the introduction to the reference but i have some notes and corrections in the attached file.

Reviewer 3 Report

In the present manuscript, Zhang and colleagues developed high-throughput screening assay for β-xylosidase enzyme from mutant strains. After screening, the authors obtained the enzyme with high β-xylosidase activity. The authors biochemically characterized the enzyme and the enzyme showed biofuctional activities (β-xylosidase and ɑ-L-arabinofuranosidase activities). In addition, the authors found that supplementation of this enzyme to commercial cellulase preparation could increase glucose and xylose yields and, in turn, increased saccharification efficiency of delignified corn stover.

            The authors performed a lot of work. However, in my opinion, this work needs much improvement because there are several parts that the authors performed with no reasons. For example, why the authors need to study 3-d structure of the enzyme. Does the information help predict the mode of action or/and mechanism of this enzyme?

            Also, the authors tried to develop the screening method for β-xylosidase. However, there are no negative and positive controls for the experiment.

            Surprisingly, the enzyme has no activity against polysaccharide xylan, but it is able to degrade xylan in pretreated biomass and enhance glucose and xylose release. Please explain this phenomenon.

Abstract

  •  

Introduction

  • More detail about β-xylosidases should be provided. The information should contain: Cazy family and its category, mode of action of the enzyme.
  • More information on multifunctional activities of β-xylosidase enzymes produced by different microbial strains should be provided.
  • Please provide more detail of the commonly used screening assays/methods of β-xylosidase activities. Please indicate pros and cons/limitation of the current methods by giving some examples. Give us reasons why the authors need to develop new methods, possibly giving potential adventages of the developed methods over those current ones.

Materials and methods

  • L73: for subdivision 2.2., I think that a brief summary of the mutaion methods, e.g., conditions used, should be described.
  • L78: What are the positive and negative controls used for this assay.
  • L131: Please indicated the company name, product version, protein content, and enzymatic activity for the commercial cellulase.
  • Please indicate the definition of the enzyme activities for β-xylosidase and ɑ-L-arabinofuranosidase activities.

Results and discussion

  • L156-159: Why the authors need show this relationship? What samples did the author used to calculate the enzyme activity (Fig. 1B)? If the authors used the fluorescence intensity for the calculation, what the authors intend to describe this result? Please explain in more detail.
  • L 184: “ piceum” should be written italic.
  • L186-189: Why did the authors need to perform this experiment (cellulase determination)? Is it related to the screening of β-xylosidase?
  • Why did the authors selected the second peak for further purification, as the activity was not determined for β-xylosidase? How did the author make a decision?
  • L223-224: Please indicate who perform genemic sequence. Is the genomic information available online?
  • Incubation of enzyme for 30 min is enough for thermal stability determination? I think incubation time should increase.
  • L309: Please explain the rationale of this study. Your enzyme prefer oligomers for hydrolysis but could degrade xylan present in the delignified substrate.

Figure

  • 1B: Please improve the Y-axis label. The text cannot be read.
  • 3A: There are no lebels indicating peak I, II, amd III.

Reviewer 4 Report

The paper studies fungal xylosidase and its role into delignification process. The soundness of the paper is good, but it needs significant improvement before publication.

line 55-62: this section should be revised. It is quite confusing and redundant. For instance, it is missing which are the common methods used to investigate xylosidases production by fungi. Then you could describe the forefront technologies, but showing why there is a still room for improvement, and why ultrahigh-throughput screening system may help.

Why your system is ultrahigh-throughput screening system? What make it Ultra compared to a high-throughput screening?

Besides, xylosidase can be produced by bacteria too. Authors do not talk about the microbial origin of these enzymes. This may be not the center of this paper, but should not be overseen in the Introduction. Authors may also state why they focus the attention of the fungal enzymes.

line 70: additional information about the aim of performing mutagenesis should be provided. Moreover additional information about P. piceum H16 is needed too. Where it is preserved? Why was it chosen? Which are the expected effect on the fungus exposed to chemical mutagenesis?

line 71: Information about the growing conditions should be provided. Gao et al 2017 citation is useless. They talk about cloning esterase DNA of Penicillium piceum and expressed in E. coli. The paper uses Luria Bertani medium for E. coli.

line 101: even though authors correctly named the reference paper, I suggest including brief information of the method used.

line 131-132: additional information about T. reseei and its enzymes should be provided.

In general, the order of presentation of methods do not follow the order of results description. Please revise this section.

line 148-154: is this a speculation? Authors should better show data that sustain these affirmations.

lie 223: P. piceum-sequenced genomic data. Please include few details about the reference genomic data used.

Table 3, statistical elaboration of data should be done.

line 312-318: It’s difficult to me to contextualize this comment with the above presented data.

line 320-321: this should be demonstrated.

In general, discussion is quite poor along the paper. The comment/discussion with literature data should be enhanced.

Reference list should be revised.

some minor comments

line 40-41: no space needed.

line 101: IEMBO not in capital letters

be consistent using ml or mL, U or IU.

always include a space between number and unit.

fungal species is always in italic

fig. 4 caption, please correct it.

table format should be revised

fig 5c-d is difficult to read.

Round 2

Reviewer 3 Report

Dear Editor,

The present manuscript was much improved, and it is useful for readers.

This version can be accepted for publication.

Reviewer 4 Report

I thanks the authors for the revision of the paper. It sounds good to me now.

Just pay attention to avoid mistyping and format minor mistakes in the revised paper.